# Biopolymer Materials in Triboelectric Nanogenerators: A Review

**DOI:** 10.3390/polym16101304

**Published:** 2024-05-07

**Authors:** Qiliang Zhu, Enqi Sun, Zequan Zhao, Tong Wu, Shuchang Meng, Zimeng Ma, Muhammad Shoaib, Hafeez Ur Rehman, Xia Cao, Ning Wang

**Affiliations:** 1Center for Green Innovation, School of Mathematics and Physics, University of Science and Technology Beijing, Beijing 100083, China; d202110424@xs.ustb.edu.cn (Q.Z.); d202310453@xs.ustb.edu.cn (E.S.); m202110789@xs.ustb.edu.cn (Z.Z.); m202210764@xs.ustb.edu.cn (S.M.); m202210762@xs.ustb.edu.cn (Z.M.); d202361034@xs.ustb.edu.cn (M.S.); hafeezurrehmandgk59@gmail.com (H.U.R.); 2National Institute of Metrology, Beijing 100029, China; wut@nim.ac.cn; 3Beijing Institute of Nanoenergy and Nanosystems, Chinese Academy of Sciences, Beijing 100083, China; 4School of Chemistry and Biological Engineering, University of Science and Technology Beijing, Beijing 100083, China

**Keywords:** biopolymers, triboelectric nanogenerator, energy harvester, self-powered sensor

## Abstract

In advancing the transition of the energy sector toward heightened sustainability and environmental friendliness, biopolymers have emerged as key elements in the construction of triboelectric nanogenerators (TENGs) due to their renewable sources and excellent biodegradability. The development of these TENG devices is of significant importance to the next generation of renewable and sustainable energy technologies based on carbon-neutral materials. This paper introduces the working principles, material sources, and wide-ranging applications of biopolymer-based triboelectric nanogenerators (BP-TENGs). It focuses on the various categories of biopolymers, ranging from natural sources to microbial and chemical synthesis, showcasing their significant potential in enhancing TENG performance and expanding their application scope, while emphasizing their notable advantages in biocompatibility and environmental sustainability. To gain deeper insights into future trends, we discuss the practical applications of BP-TENG in different fields, categorizing them into energy harvesting, healthcare, and environmental monitoring. Finally, the paper reveals the shortcomings, challenges, and possible solutions of BP-TENG, aiming to promote the advancement and application of biopolymer-based TENG technology. We hope this review will inspire the further development of BP-TENG towards more efficient energy conversion and broader applications.

## 1. Introduction

With the escalating demand for clean and alternative energy sources, researchers have investigated various technologies, including photovoltaic systems [1], piezoelectric devices [2], and thermoelectric transducers [3], aimed at converting diverse renewable energy forms from the ambient environment into electrical power. Among these, mechanical energy is considered to be abundant and omnipresent in daily life [4,5]. The triboelectric nanogenerator (TENG), an emergent technology in energy harvesting, demonstrates the capability to ubiquitously transform environmental mechanical energy into electrical power across various contexts and temporal settings [6,7,8,9,10]. This includes applications in flow-driven triboelectric generators for the autonomous powering of wireless sensor nodes [11,12,13], as well as in triboelectric generators designed for the simultaneous collection of diverse energy forms [14,15,16,17]. Their applications are not limited to power generation, as they also realize self-powered sensing by improving device structures, rough surfaces, and the incorporation of nanomaterials; for instance, arrays of TENG enhanced with electret films have been developed for self-powered, instantaneous tactile imaging [18], and temperature-sensitive TENGs [19,20]. However, most research efforts focus on studying and developing high-performance TENGs, with little attention given to renewable materials or devices. For example, the polymers most widely used in TENGs, such as polypropylene [21], polyvinyl chloride [22], polystyrene [23], nylon [24], Teflon [25], and polyurethane (PU) [26], usually come from petroleum, are not fully degradable, and can release harmful chemicals into the environment. Therefore, there is significant interest in using biodegradable, renewable, and easily prepared biopolymer materials for the sustainable and proper application of TENG [27,28,29].

Currently, researchers have explored a range of biopolymers such as cellulose, lignin, chitin, chitosan (CHS), fish gelatin (FG), silk fibroin (SF), and alginate (Alg), which have been selected as core components for TENGs due to their unique chemical and physical properties. In particular, polysaccharides like cellulose and chitin [30,31], as well as certain specific proteins such as collagen, keratin, and SF secreted by organisms like silkworms and spiders, have demonstrated complex structural patterns and biological functionality, providing new directions for the design and functionality of TENGs [32]. These carbon-neutral biopolymers from plants and animals offer advantages unattainable by petroleum-based materials [33]. For example, Kim et al.’s study showcased how SF could be transformed into films through electrostatic spinning techniques [34], further used as the triboelectric layer in BP-TENG, demonstrating not only the direct application of biopolymers in energy applications but also highlighting their advantages in terms of environmental friendliness and renewability. The use of these materials has facilitated the realization of a variety of self-powered applications, especially showing remarkable performance in the field of self-powered medical and diagnostic sectors. From smart electronic devices to health monitoring systems, BP-TENGs will influence our future way of life.

This paper delves into the working principles, material selection, and application fields of BP-TENG, showcasing a wide range of categories from natural sources to microbial synthesis and chemically modified biopolymers. The application of these biopolymers not only demonstrates advantages in biocompatibility and environmental sustainability but also shows great potential in enhancing TENG performance and expanding application scenarios. Additionally, the paper reveals the challenges and future prospects faced by the development of BP-TENG, emphasizing the importance of interdisciplinary collaboration in advancing biopolymer-based TENGs towards more efficient energy conversion and broader application fields (Figure 1).

## 2. Basic Principle and Working Modes of TENGs

In 2012, Zhong Lin Wang’s team introduced the world’s first TENG. Since then, applications based on TENG have been emerging in various fields, offering new possibilities for harvesting and reusing physical or physiological energy that was previously challenging to utilize. This discovery has greatly sparked researchers’ enthusiasm for further exploration [45]. This technology leverages the triboelectric effect and electrostatic induction, two ancient and familiar natural phenomena, to provide clean energy solutions for various applications [46]. Essentially, when two different materials come into contact, the transfer of electrons between them generates charges. These charges lead to the generation of a potential difference when the two objects separate, which forms the basis for TENG’s electricity generation. Through repetitive contact and separation, along with the relative motion between electrodes, TENG can produce periodic electrical current pulses, powering a variety of devices [47]. Over time, researchers have progressively developed and refined the four principal operational modes of the TENG, grounded on the principle of contact electrification. These include the vertical contact/separation mode, lateral sliding mode, single-electrode mode, and freestanding triboelectric layer mode [48,49].

In the vertical contact/separation mode, the TENG consists of two different triboelectric material layers, which contact and separate from each other in the vertical direction (Figure 2a). When these two layers come into contact, charges are exchanged between the two contact surfaces due to friction, and a potential difference is generated as they separate, causing a current to flow through the connected external circuit. This cyclical contact and separation process produces alternating current [50,51].

The lateral sliding mode is similar to the vertical contact/separation mode, but the difference lies in the relative movement of the triboelectric layers, which occurs in a parallel direction (Figure 2b). This leads to periodic changes in charge related to changes in the contact area, thereby generating a potential difference and current [52,53].

The single-electrode mode has a simpler structure, where only one triboelectric layer is associated with a grounded reference electrode (Figure 2c). In this mode, the moving object contacts the triboelectric layer to generate charges, and the change in potential causes a current to flow towards the grounded reference electrode. This mode allows for more freedom of object movement and is suitable for irregular or random motions [54,55].

The freestanding triboelectric layer mode includes a freely movable triboelectric layer corresponding to two fixed electrodes (Figure 2d). The movement of the object changes its relative position to the electrodes, thereby generating a potential difference and current. This mode is particularly suitable for applications requiring higher device integration and durability [56,57].

Each mode has its unique advantages and application scenarios, allowing TENG to flexibly harvest energy according to different environments and needs. The comprehension and implementation of these operational modes have significantly advanced the development of TENG technology within the realms of energy harvesting and autonomous devices.

## 3. Recent Progress in BP-TENGs

### 3.1. BP-TENG Based on Natural Polymers

Most TENGs are constructed based on synthetic polymers that lack biocompatibility, limiting their applications in biomedicine and implant scenarios. Employing naturally abundant materials, previously deemed as waste, for the harvesting of unutilized mechanical energy, is pivotal not only for fundamental scientific inquiry but also for meeting practical societal demands. When selecting materials for TENGs, researchers need to consider the biocompatibility, low cost, flexibility, and durability of the materials.

#### 3.1.1. Polysaccharide-Based BP

To achieve a greener environment, polysaccharide-based biopolymers such as cellulose [58], lignin [59], and starch [60] have been utilized as the dielectric materials for BP-TENGs.

For example, cellulose, sourced from wood and a variety of green plants, represents one of the Earth’s most prevalent organic polymers. Owing to its rich oxygen atom content, cellulose manifests distinct physicochemical characteristics that promote electron detachment. Therefore, in previously reported studies, cellulose was often chosen as the friction-positive material for eco-TENGs [61]. Yao et al. were pioneers in introducing cellulose-based eco-TENGs [35]. They constructed a TENG device by combining a flexible, transparent cellulose nanofiber (CNF) film with fluorinated ethylene propylene (FEP), resulting in a performance that rivals that of synthetic polymers, as depicted in Figure 3a. Additionally, the CNF-based TENG was integrated into a fibrous board made from recycled cardboard fibers using a chemical-free cold-pressing method. This fibrous board is capable of generating up to 30 V and 90 μA under human footstep, as shown in Figure 3b,c, indicating potential for large-scale and environmentally sustainable applications. Taking into account the properties of CNF and other natural wood-derived materials, this research provides new opportunities for developing eco-friendly flooring, packaging, and infrastructure with unique mechanical energy-harvesting capabilities.

Lignin, the most abundant aromatic biopolymer in nature, provides structural support to plants, enhancing their biomechanical strength. Benefiting from its biodegradability and biocompatibility, lignin offers valuable opportunities for low-cost TENG applications in biomedical devices. Nutshells, typically discarded as waste, contain a high amount of lignin. Saqib and others demonstrated the potential of lignin-based WFSs (Waste from Shells) as the friction-positive material for TENG [38]. The study focused on WFS from almonds (As), walnuts (Ws), and pistachios (Pis), with a detailed examination conducted on pistachio WFS, as illustrated in Figure 3d. TENG devices were designed by combining WFS with materials such as polytetrafluoroethylene (PTFE) and polyethylene terephthalate (PET), as shown in Figure 3e, where Pi-WFS exhibited the best output performance, including the maximal open-circuit voltage, short-circuit current, and peak power density, as highlighted in Figure 3f. These WFS-based TENGs were also successfully applied to charge commercial capacitors, illuminate LEDs, power stopwatches, and electronic calculators. This research presents a promising approach to generating electrical energy from discarded biomaterials in eco-friendly, efficient, and sustainable energy utilization systems, with broad applications in self-powered wearable electronics and the Internet of Things.

Additionally, polysaccharides contain a large number of hydroxyl (-OH) groups, which can form hydrogen bonds with water molecules, endowing polysaccharides with excellent water solubility and hydrophilicity. Starch represents an accessible, natural, edible, and biodegradable polymer. Nevertheless, the intrinsic hydrophilic property of raw starch presents challenges in the context of TENG applications. Khandelwal and colleagues developed a starch/seaweed-TENG [62]. They introduced edible seaweed as a filler to enhance the hydrophobicity of starch while maintaining its edibility. The developed TENG did not degrade in phosphate-buffered saline (PBS) for 30 days. Moreover, compared to raw starch, the TENG output increased by four and nine times (voltage and current), respectively.

#### 3.1.2. Protein-Based BP

Natural biopolymers based on proteins obtained from animals and plants, such as silk, gelatin, plant proteins, egg whites, peptides, etc., can be used as the triboelectric layer in TENGs. Due to the rich amide groups (electron-donating groups) on the protein backbone, these triboelectric materials typically exhibit strong electron transfer capabilities. Moreover, these proteins can quickly degrade in environments with microbes and proteases and possess good biocompatibility.

For example, SF is an animal-derived degradable material extracted from domestic silkworms, occupying a top position in the triboelectric series due to its abundance of carboxyl groups [63], and exhibits excellent electron donation capabilities. Given its biocompatibility and transparency in the visible region, it has broad application prospects in the biomedical field. Candido and others adopted a simple and fast method to incorporate SF into polyvinyl alcohol, creating a PVA/SF-based TENG [36]. This impregnation process affects all polarization processes and directly impacts the dielectric properties of the device. However, pure SF has not reached commercial standards in terms of mechanical properties and chemical stability; pure silk materials are prone to fracture and exhibit poor stability under high temperatures, high humidities, and strongly acidic or alkaline conditions. Xu and colleagues developed a flexible, stretchable, and fully bioabsorbable TENG for harvesting biomechanical energy outside or inside the body [64]. They introduced mesoscopic doping to promote the secondary structure transformation of regenerated SF. The doped silk film exhibited an excellent chemical stability (withstanding temperatures of 100 °C and pH values of 3–11) and outstanding mechanical stability (∼250% stretchability and 1000 bending cycles at a radius of 2 cm), making them broadly applicable in human health.

Gelatin, typically derived from the connective tissues of animals (such as fish, cattle, etc.), forms a gel under incomplete hydrolysis by proteases. Sun and others manufactured a flexible, transparent, fully sustainable, and high-performance FG-based TENG [40]. By modifying the friction layers with dopamine and fluorosilane, a triboelectric pair was formed, as illustrated in Figure 3g. This TENG demonstrated notable output characteristics, including an open-circuit voltage reaching 500 V and a short-circuit current of 4μA, accompanied by a power density of 100 μW·cm^2^, as highlighted in Figure 3h. This innovation not only provides direct or indirect power to small electronic devices but is also applicable in fields such as human motion energy harvesting and human/machine interaction, offering a new sustainable solution for green, cost-effective, and wearable electronic products, as demonstrated in Figure 3i.

Plant proteins, exemplified by rice protein (RP), commonly emerge as by-products in the starch industry and are frequently utilized as boiler fuel and animal feed, resulting in considerable resource wastage. Jiang and colleagues used recovered plant protein rice gluten (RG) from starch industry by-products as a model to study the mechanism of triboelectric charge behavior related to protein structure [65]. Through simple pH cyclic interface engineering techniques, they discovered that the secondary structure of RG significantly impacts its triboelectric performance, proposing and validating possible mechanisms. They successfully achieved a ~16-fold increase in the output power density of the BP-TENGs and were able to manipulate the triboelectric performance of proteins. This work not only made significant progress in resource recovery but also highlighted the tunable properties of proteins as soft materials, providing potential biopolymer solutions for sustainable applications in next-generation intelligent packaging, wearable technology, and implantable medical devices.

Given the diversity of protein structures and the ease of modification and improvement through biological and chemical methods, new biopolymer-based TENGs can be engineered. For instance, prior research has demonstrated the utilization of recombinant spider silk protein to augment triboelectric performance via genetic engineering techniques [66]. However, the extraction and purification processes for these biopolymers are complex and time-consuming [67,68]. Therefore, exploring new biopolymers that are simple and economical in processing and manufacturing is crucial for successful application in TENG devices.

### 3.2. BP-TENG Based on Microbial Synthetic Polymers

Under controlled conditions, microorganisms such as bacteria, fungi, and algae can serve as factories, transforming various carbon and nitrogen sources into a variety of intracellular and extracellular biopolymers. The biopolymers produced by microbial systems are rich in various functional groups, which can be further utilized to modify the polymers for use in green and environmentally friendly energy-harvesting technologies.

#### 3.2.1. Bacterial BP

Polymers synthesized by bacteria, such as polyhydroxyalkanoates (PHAs), polysaccharides, and polyamine acids, offer a sustainable option for biodegradable materials, with significant environmental benefits including reduced dependence on fossil fuels and the decreased impact of plastic waste on the environment.

Currently, scholars have identified over 150 different monomeric structures of PHAs [69]. Among these, polyhydroxybutyrate (PHB) is characterized by its simple and uniform structure, serving as a prototypical member of PHAs [70]. In environments controlled for nutrients such as nitrogen, oxygen, phosphorus, and mineral ions, certain bacteria produce large amounts of PHB internally during fermentation. For example, Halomonas boliviensis can produce PHB using carbon sources such as fructose, xylose, fumarate, itaconate, propionate, and lactate [71]. Due to its short degradation cycle, excellent biocompatibility, and good triboelectric performance, PHB holds significant advantages and potential applications in transient bioelectronic devices [72].

For instance, Wang and colleagues constructed a bio-TENG based on high-pressure crystallized polyhydroxy butyrate (HP-PHB) [70]. Compared to PHB crystallized at normal pressure (NP-PHB), HP-PHB exhibited a unique wrinkled spherulite and dual-scale crystalline structure, significantly improving its triboelectric generation performance. The voltage and current outputs of HP-PHB-based single-electrode TENGs were found to be approximately fivefold and twelvefold greater than those of NP-PHB-based TENGs, respectively, achieving peak values of 25.6 V cm^−2^ and 550.2 nA cm^−2^. Additionally, the study clarified the intrinsic relationship between the multiscale dense state structure of HP-PHB and the surface charge transfer and output performance of BP-TENGs. This research offers a new perspective for designing and manufacturing high-performance TENGs and environmentally friendly transient energy devices.

Bacterial cellulose (BC), a natural nanofiber polymer material belonging to the polysaccharide category, is mainly produced by bacteria such as Gluconacetobacter xylinus from the Acetobacter genus through a biosynthetic pathway. It is biodegradable, and Gluconacetobacter is considered one of the strongest microbial producers of cellulose. Its unique physical and chemical properties, such as high strength, high purity, good biocompatibility, and biodegradability, make it an excellent material for flexible electronics and TENGs. For example, Zhang and others developed an environmentally friendly, biodegradable energy-harvesting and interaction device based on BC suitable for TENG [73]. Enzymatic hydrolysis experiments proved that the active materials in the device could completely degrade within 8 h and demonstrated good electrical energy output performance. The TENG could not only power commercial electronic devices but also serve as a wearable sewn interface to control an electronic piano. This study provides a new method for fabricating degradable energy-harvesting devices, contributing to the advancement of eco-friendly electronics and wearable devices.

As shown in Figure 4a, Chen and colleagues introduced a novel environmentally friendly superhydrophobic fabric-based triboelectric nanogenerator (SF-TENG) [39], woven from superhydrophobic electroconductive bacterial cellulose (SEBC fibers). To construct durable superhydrophobicity, an ingenious biomimetic method was employed to form a shell/core structure. The SEBC fibers with the biomimetic shell/core structure exhibited excellent electrical conductivity, mechanical properties, biodegradability, and enduring superhydrophobicity. The SF-TENG demonstrated a maximum open-circuit voltage of 266.0 V, a short-circuit current of 5.9 µA, and an output power of 489.7 µW, as depicted in Figure 4b, successfully powering devices such as stopwatches and calculators, as shown in Figure 4c. Its self-cleaning and anti-contamination capabilities ensured the stable output performance of the SF-TENG under harsh environmental conditions such as liquid spillage. This provides a new biomimetic manufacturing strategy for the design and preparation of superhydrophobic conductive fibers and demonstrates the practicality and stability of the SF-TENG in adverse environmental conditions.

The direct application of polyamine acids in the TENG field is relatively rare. Typically, polyamine acids can be used as materials for TENGs’ power-generating layers by introducing specific functional groups or combining them with other conductive materials (such as metal nanoparticles, conductive polymers, etc.) to improve their conductivity and triboelectric properties. For example, Suktep and others developed a hybrid organic piezoelectric/triboelectric nanogenerator (HO-P/TENG) using natural SF (SF) and γ-glycine (γ-gly) composite films [74]. By altering the content of γ-gly, optimal conditions were identified to achieve maximum output efficiency. At a 15% γ-gly concentration, the composite film showed the highest output voltage and current (V_OC_ of 81 V, I_SC_ of 121 μA). This HO-P/TENG device, due to its cost-effectiveness, simple preparation process, and good performance characteristics, demonstrated a great potential for future electronic energy-harvesting devices.

Khandelwal and colleagues researched and developed one-dimensional nanofibers constructed from aspartic acid and copper ions (Cu-Asp NFs) for creating a new type of TENG [75]. Utilizing a straightforward coating method, these nanofibers demonstrated superior stability and adhesion. Furthermore, improved ionic deposition techniques significantly enhanced the electrical performance of the TENG. The research was also successfully applied to self-powered sensors and powering portable electronic devices, showcasing the potential application value of aspartic acid in energy-harvesting fields.
Figure 4(**a**) Morphology of the SEBC fibers; (**b**) the contact angles of SEBC fibers with water, HCl, NaOH, NaCl, and PBS solutions; (**c**) output performance of the SF-TENG; reprinted with permission from Ref. [39]. Copyright 2023, Wiley Online Library. (**d**) The schematic diagram of the TENG device based on the BSF biolayer/PTFE structure; (**e**) the alpha and beta structures of CHS; (**f**) the short-circuit current and power of the TENG; reprinted with permission from Ref. [76]. Copyright 2023, Wiley Online Library. (**g**) TENG based on SA and Ag NWs/SA; (**h**) the analysis of peak power and voltage in the SA-TENG across varying resistances; (**i**) the evaluation of degradability and antibacterial efficacy in the SA-TENG apparatus. Reprinted with permission from Ref. [77]. Copyright 2023, Elsevier.
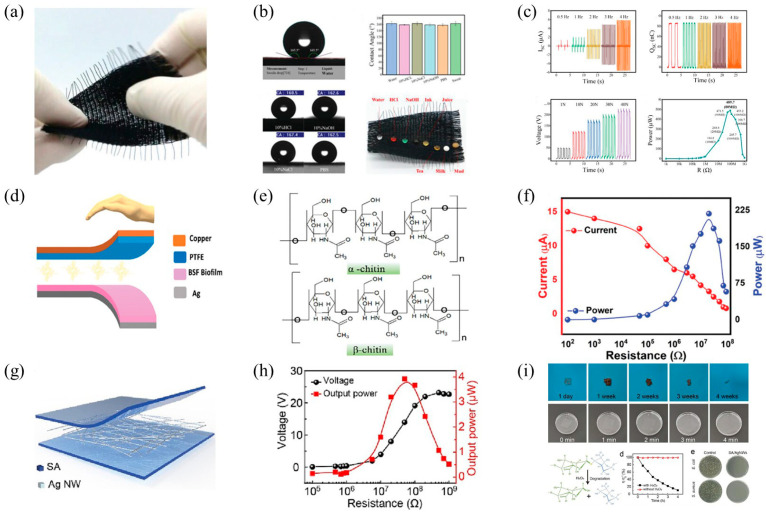


#### 3.2.2. Fungal BP

Fungi, as widely distributed microorganisms on Earth, are not only vital for the balance of ecosystems but also important biological resources for producing various useful polymers. Chitin is one of the most typical polymers synthesized by fungi, being a major component of fungal cell walls. The presence of hydroxyl, carbonyl, α-chitin, and β-chitin functional groups endows chitin with good electron-donating capabilities. Moreover, the presence of non-centrosymmetric β-sheets describes the piezoelectric characteristics of chitin. These properties pave the way for chitin’s applicability in advanced energy-harvesting and sensing technologies.

For instance, Patil and colleagues proposed a CHS biofilm derived from black soldier fly (BSF), used to manufacture high-performance triboelectric and piezoelectric nanogenerators (Figure 4d) [76]. Through a detailed physicochemical analysis, it was demonstrated that the CHS biolayer exhibits significant positive triboelectricity and piezoelectric properties due to the presence of hydroxyl groups, α-CHS, and β-CHS, as illustrated in Figure 4e. Experimental results showed that the TENG based on BSF/PTFE could generate a voltage of 121 V and a current of 15 µA, as depicted in Figure 4f. According to the results presented, CHS derived from BSF sources is a strong candidate for developing flexible, environmentally friendly energy-harvesting and self-powered energy storage systems.

Petchnui and others utilized chitosan nanofibers (ChNFs) extracted from shrimp shells mixed with natural rubber (NRL) to create an environmentally friendly TENG material [78]. ChNF was extracted from shrimp shells using a simple mechanical method and incorporated into NRL to form an NRL/ChNF composite material. This composite material not only displayed improved mechanical and dielectric properties but also significantly increased the output voltage to 106.04 ± 2.3 V in TENG applications, demonstrating excellent energy conversion efficiency. This work showcases an environmentally friendly and efficient method of transforming biological waste into high-performance materials for energy-harvesting devices.

Overall, polymers synthesized by fungi play multiple roles in nature and human society. They are not only fundamental for the survival of fungi but also precious resources for modern biotechnology and industry. With further research into fungal biology and the development of bioengineering techniques, the potential for developing and utilizing novel polymers from fungi remains immense.

#### 3.2.3. Algae BP

Algae, as a bioresource with rapid growth characteristics and environmental friendliness, have been proven to be a platform with enormous potential for biopolymer production. These organisms not only can perform photosynthesis using CO_2_ as a carbon source, thereby providing oxygen for the planet, but their growth does not consume food resources or require arable land and freshwater. Algae can synthesize a variety of polymers through biotransformation processes and are widely used in multiple fields due to their renewable source, biodegradability, and biocompatibility, including food, pharmaceuticals, agriculture, water treatment, and biomedicine. The main polymers synthesized by algae include Alg, agar, and carrageenan, and their production is rapid, providing potential alternatives for different polymers such as PHA.

Sodium alginate (SA), a polysaccharide copolymer derived from brown seaweeds, has gained increasing importance due to the over-exploitation of terrestrial resources and the advancement of global marine resource utilization. SA is composed of (1–4)-linked β-D-mannuronic acid and α-L-guluronic acid and is abundant in hydroxyl groups, which serve as electron donors, rendering it an effective positive triboelectric material for TENG applications [79]. Furthermore, SA’s excellent film-forming capabilities and the high degree of transparency render it an ideal material for the development of flexible, transparent, and transient wearable electronic devices.

Li and colleagues reported a biodegradable, transparent, and antibacterial SA-based TENG for mechanical energy harvesting and self-powered tactile sensing (Figure 4g) [80]. The flexibility and adhesiveness of the SA were enhanced by adding glycerol, rendering the Ag NWs/SA electrodes highly transparent and conductive. The output voltage, transferred charge, and peak power of the TENG were measured to be 53 V, 18 nC, and 4 μW, respectively, demonstrating adequate capacity to energize small electronic devices, as depicted in Figure 4h. Moreover, this TENG also demonstrated good antibacterial and biodegradable properties, as illustrated in Figure 4i. This work successfully enhanced the performance and eco-friendly characteristics of TENG by employing SA/glycerol composite films and Ag NWs, offering new strategies for developing sustainable self-powered devices and transient electronic products.

Similarly, Xia and others proposed a novel multifunctional triboelectric nanogenerator (AMC-TENG) based on alginate-metal complexes, tuning electrical output through Alg compounded with different metal ions [81]. Experimental results showed that AMC-TENG exhibited excellent electrical output, especially the output from alginate-copper (Alg-Cu). Moreover, the study also developed a self-powered sensing array, showcasing its potential in practical applications. This research contributes novel insights and methodologies to the study of triboelectric materials and their implementation in autonomous sensing systems.

κ-Carrageenan and agar are naturally sourced, cost-effective, edible polysaccharides, primarily composed of galactose linked by secondary bonds [82]. They are typically extracted from red seaweeds or other plants, displaying high intrinsic biocompatibility [83]. Moreover, they are degraded and absorbed into the body through microbial hydrolysis or enzymatic hydrolysis. κ-Carrageenan-agar composites have potential as excellent triboelectric materials; for instance, Kang and colleagues developed a biodegradable composite made of carrageenan and agar, used as materials for effective biomechanical energy harvesting [77]. These polysaccharides extracted from red algae, by forming a three-dimensional network structure, significantly enhanced the triboelectric performance. The composite possesses high biocompatibility and degradability, suitable for in-body applications, and the TENG it produced could power LEDs and capacitors without an external power source, demonstrating its potential application in the biomedical field.

### 3.3. BP-TENG Based on Chemically Synthesized/Modified Biopolymers

The process of synthesizing polymers in the laboratory involves various chemical methods, covering ring-opening polymerization (ROP), condensation reactions, free radical polymerization, copolymerization, and enzyme-catalyzed reactions, as well as chemical modifications. Although some of these methods are also used for the synthesis of non-biological polymers, here we focus specifically on those pathways capable of producing biopolymers.

#### 3.3.1. Ring-Opening Polymerization

ROP is a technique commonly used for synthesizing biopolymers, primarily involving the polymerization of cyclic monomers. Biopolymers synthesized through this method mainly include polylactic acid (PLA) and polycaprolactone (PCL).

PLA is one of the most promising biobased polyesters for food packaging. It is a biocompatible thermoplastic with a melting temperature of 175 °C and a glass transition temperature of 60 °C, produced as a byproduct of corn wet milling during the fermentation process of corn starch. It has good mechanical properties, transparency, ease of processing, and barrier properties. However, its use in food packaging is somewhat limited due to poor ductility, thermal barrier, and oxygen barrier properties compared to starch and cellulose [84]. Compared to starch and cellulose, PLA is a highly durable biobased polymer.

Zhao and colleagues engineered a helix-shaped triboelectric nanogenerator (H-TENG) utilizing PLA [85], aimed at effective energy collection, as depicted in Figure 5a. Fabricated through 3D printing techniques, this apparatus can, upon manual compression, deliver an impressive open-circuit voltage reaching 395 V and a short-circuit current of 28 μA, powerful enough to illuminate 300 LED lights, as demonstrated in Figure 5b. Exhibiting high energy conversion efficiency, the H-TENG is capable of autonomously powering devices such as digital wristwatches and calculators, as showcased in Figure 5c. Its outstanding capability to gather energy and superior output make it a promising candidate for use in portable electronic gadgets and interactive technologies between humans and machines.

PCL is a biodegradable polyester synthesized through ROP reactions. PCL is anticipated for TENG technology due to its good biocompatibility, biodegradability, and mechanical properties. For instance, Luo and colleagues developed a novel self-healing TENG utilizing a combination of PCL and silver nanowires (Ag NWs) [86], enhancing the durability of TENGs in long-term energy harvesting. The study found that softening the PCL by heating could effectively repair damage to the TENG’s friction surface and the conductive layer underneath. After several cut/heal cycles, the TENG maintained stable high output performance, reaching an output voltage of 800 V and short-circuit current of 30 μA, capable of lighting 372 LEDs. Moreover, the TENG was also modified into a self-powered vibration sensor for real-time monitoring and diagnosing mechanical equipment malfunctions. This work provides effective strategies for extending the lifespan and practical application of TENGs.

Additionally, Li and colleagues proposed a novel temperature-responsive liquid/solid TENG based on PCL, achieving tunable triboelectricity [87]. The structure of the PCL was adjusted by changing temperatures, thereby controlling the triboelectric effect between the liquid and solid phases. This PCL-based TENG showed significant electrical output changes with temperature variations, proving its effectiveness as a real-time monitoring interface state. Furthermore, the study also explored the reversibility and stability of PCL material, offering new control strategies for triboelectric performance at liquid/solid interfaces.
Figure 5(**a**) Schematic representation of the H−TENG architecture; (**b**) the electrical output performance of the H−TENG; (**c**) electronic devices powered by the H−TENG; reprinted with permission from Ref. [85]. Copyright 2023, Elsevier. (**d**) Flowchart for the synthesis of CD−MOF-based TENG via the ultrasonic method; (**e**) the comparison of output voltages for A-TENG, B-TENG, and G-TENG; (**f**) CD−MOF-based TENG powering various low−power electronics; reprinted with permission from Ref. [88]. Copyright 2021, Wiley Online Library. (**g**) The schematic illustration of the TENG configuration utilizing PDMS and CANF; (**h**) the output power of the TENG with pyramid-shaped microstructures under different load resistances; (**i**) the schematic diagram and sensing curve of the electric sewing machine device. Reprinted with permission from Ref. [89]. Copyright 2022, Elsevier.
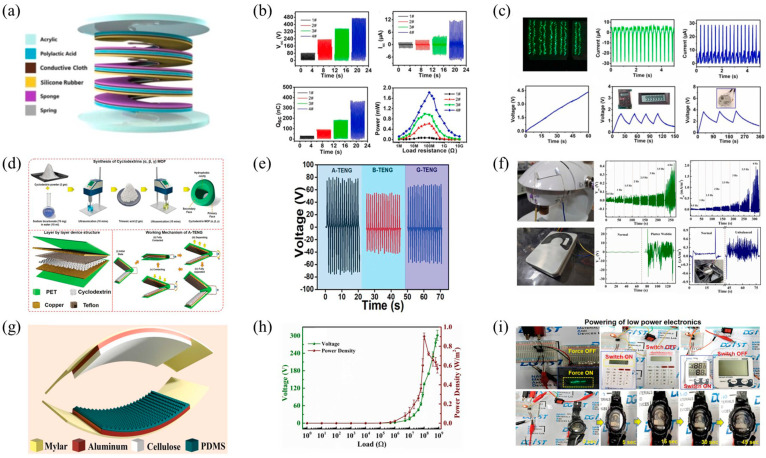


#### 3.3.2. Enzyme-Catalyzed Polymerization

This is a newer method where enzymes catalyze the polymerization of monomers. For example, CHS can be synthesized from chitin through enzymatic means. This process involves using specific enzymes, such as chitinase, to remove acetyl groups from chitin.

CHS is an abundant natural biopolymer derived from the shells of marine crustaceans, offering exciting opportunities for low-cost, biocompatible TENG applications [90]. In terms of electrostatic properties, CHS tends to lose electrons as a positively charged material, degrades into smaller molecules in the environment, and degrades slowly in the human body, exhibiting good biocompatibility and bioabsorbability. Additionally, CHS has excellent antimicrobial action and impressive gelation properties, making it an exemplary material with broad applications. For instance, Menge and others prepared biopolymer films based on CHS and Alg through layer-by-layer self-assembly methods, used to construct an antibacterial TENG [91]. The TENG showed the highest electrical output performance with voltages reaching 474 V and current densities of 36.9 mA/m^2^, attributed to the highest surface potential and lowest work function, respectively, at 239.4 mV and 4.2 eV. This novel TENG demonstrates application potential in health monitoring and portable electronic devices due to its low cost, environmental friendliness, and excellent electrical output.

Cyclodextrins (CDs) are a class of cyclic polysaccharides formed by glucose units linked by α-1,4-glycosidic bonds. Their cyclic structure grants CDs unique physical and chemical properties, especially their hydrophobic inner cavities and hydrophilic outer surfaces, enabling them to form inclusion complexes by embedding various small molecule guests into their hydrophobic cavities. β-CD is obtained from starch through the action of microbial enzymes like cyclodextrin glycosyltransferase (CGTase), which can break down and rearrange starch into CD. CDs are frequently employed to enhance the electrical output of TENG. In a notable example, Hajra et al. fabricated a metal/organic framework (MOF), employing sodium as the metal ion and CD as the organic ligand, referred to as CD-MOF. This was achieved using an ultrasonic synthesis approach, specifically for integration within TENGs [88], as represented in Figure 5d. The output performance of various CD-MOF-based TENGs varied, with the hierarchy being α-CD MOF/Teflon > γ-CD MOF/Teflon > β-CD MOF/Teflon, as illustrated in Figure 5e. The devices produced were capable of powering multiple low-power electronic devices through capacitors and bridge rectifiers, as shown in Figure 5f. Utilizing biocompatible CD-MOFs (α, β, γ) for energy harvesting and triboelectric series marks a novel research direction in the realms of environmental sustainability and self-powered technologies.

#### 3.3.3. Chemical Modification

Cellulose acetate (CA) is prepared through the chemical reaction of modifying natural polymers, classified as a type of esterification reaction. In this process, natural cellulose reacts with acetic anhydride, and in the presence of a catalyst, the hydroxyl groups of cellulose are substituted by acetyl groups to form CA. Due to its non-toxicity, non-irritability, good processability, film-forming ability, cost-effectiveness, biodegradability, and biocompatibility, it is considered an attractive material for triboelectric applications. Typically, CA biomaterials are spun into electrospun nanofiber films using electrospinning technology for constructing green wearable TENGs.

Bai and others developed a nanofiber composite material consisting of cellulose acetate (CA) and carbon nanotubes through electrospinning technology [92]. This material not only exhibits multi-responsive shape memory capabilities but also functions as a self-powered pressure sensor. The application of electrospinning technology allowed the CA nanofiber composite material to exhibit excellent mechanical and thermal performance while also having a heavy load capacity and high-sensitivity sensing performance. The study of CA nanofiber composite materials with multi-responsive shape memory and self-powered sensing performance significantly broadens its application prospects.

Similarly, Varghese et al. explored a TENG based on CA nanofibers and surface-modified PDMS [89], depicted in Figure 5g, designed to power small electronic devices and serve as a self-powered vibration sensor. By incorporating microstructures such as microcones and microdomes into the PDMS surface, they significantly enhanced the output performance of the TENG, with a power density increase of approximately 180 times, as demonstrated in Figure 5h. Moreover, the authors transformed this TENG into a self-powered sensor capable of detecting mechanical vibrations, successfully monitoring the vibration patterns of devices such as electric sewing machines, and predicting malfunctions in computer fans and hard drives ahead of time, as shown in Figure 5i. This work illustrates the potential of CA nanofibers in enhancing energy collection and sensing technologies.

PU is typically not classified as a biopolymer since most are synthesized chemically from petrochemicals. However, PU can also be “bio-based,” meaning some or all of its components come from biological resources, such as soybean oil or castor oil. PU elastomers, known for their excellent mechanical properties, aging resistance, good compatibility, and easy modification, are also hotspots in scientific research, especially in the TENG field. For example, Cheng and others significantly improved the self-healing ability and durability of a TENG by incorporating disulfide bonds and metal coordination bonds into PU [93]. The PU elastomer demonstrated a self-healing efficiency of 85.5% and notable toughness, facilitating the TENG’s attainment of a short-circuit current of 12 μA, an open-circuit voltage of 120 V across a 2 cm × 2 cm area, and a power density of 2.1 W·m^–2^. Even after self-healing, its electrical performance remained at 95%. This research advances the application of PU in human/machine interaction and self-powered sensing fields.

Cellulose paper, as a material for electronic devices, has been used to construct TENGs, offering advantages such as low cost, lightweight, flexibility, environmental friendliness, and disposability. However, traditional papers utilized in fabricating TENGs, including cardstock, printing paper, cardboard, rice paper, and crepe paper, are commercially available but lack antibacterial properties, constraining the utility of P-TENGs due to their reduced flexibility and the absence of antibacterial characteristics. Furthermore, the surface modification of cellulose fibers, essential for the preparation of cellulose paper, is frequently neglected in the P-TENG manufacturing process. Consequently, the fabrication of the P-TENGs employing composite paper made from surface-modified cellulose fibers is essential.

For example, Piwbang et al. proposed a method to enhance the power output of TENG by the dye modification of cellulose paper [94]. The dyes used include chlorophyll, anthocyanin, and curcumin, among which the chlorophyll-modified TENG demonstrated the best performance, with a power density reaching up to 3.3 W/m^2^. This superior performance is attributed not only to the photosensitivity of the dyes but also to the molecular structure of the dyes, which promotes the electron-donating properties of cellulose.

Through these methods, researchers can design and synthesize an array of different biopolymers that not only play crucial roles in medical, drug delivery, food industry, and environmental sciences but also provide new directions and possibilities for sustainable development and green chemistry. As research deepens and technologies advance, these synthetic pathways will continue to offer innovative materials and solutions to meet societal needs and challenges. 

## 4. Possible Applications of BP-TENG

### 4.1. Energy Harvesting

BP-TENG, as an innovative energy-harvesting technology, is opening new chapters in the field of sustainable energy. It combines the green attributes of biopolymers with the efficient energy conversion capabilities of TENG, offering an eco-friendly way to harvest energy from daily activities and natural phenomena. This unique energy-harvesting technology not only reduces reliance on traditional energy sources but also heralds the arrival of a new era that can harmonize more closely with the natural environment while meeting modern technological demands.

For example, Dai et al. engineered a crack-effect-based triboelectric nanogenerator (CE-TENG) optimized for the efficient harvesting of wind energy, which was subsequently applied in autonomous wind direction and speed monitoring systems (Figure 6a) [95]. By using transparent degradable hydroxyethyl cellulose films, the CE-TENG could generate an output voltage of up to 600 V at a wind speed of 7 m/s, as shown in Figure 6b. The paper also presents an omnidirectional wind energy harvester (OWEH), composed of eight CE-TENG units, capable of accurately monitoring wind speeds and directions ranging from 0.5 to 10 m/s, as illustrated in Figure 6c. This system not only significantly enhances the sensitivity of wind speed monitoring but also supplies its own power to drive agricultural sensors, offering an effective and green approach to wind energy utilization and monitoring for smart agriculture, demonstrating the potential of converting wind energy into a sustainable energy supply.

As depicted in Figure 6d, Ding and others developed a green wrinkled paper-based TENG device for collecting and converting water wave energy (i.e., blue energy) [41]. By adjusting the wavelength and amplitude of the wrinkled paper and utilizing metal balls rolling inside the channels of a grid device (G-TENG) with the motion of the waves, the device efficiently converts water wave energy into kinetic energy, as shown in Figure 6e. Additionally, by employing superhydrophobic cellulose micro/nanostructures and biodegradable materials produced through electrospinning, the contact area and triboelectric charge density were enhanced, thus improving the efficiency of the charge transfer, as illustrated in Figure 6f. Compared to flat structures, this wrinkled structure’s TENG unit demonstrates a better adaptability and output performance in converting water wave energy. This study provides new insights into improving the efficiency of sustainable TENG arrays in collecting and converting water wave energy.

### 4.2. Medical and Health

#### 4.2.1. Human Health Diagnosis

Breathing is not only a fundamental sign of life but also an important reflection of metabolic status and health condition. By analyzing the respiratory signals or biomarkers in a breath, such as acetone, nitric oxide, and carbon dioxide, various diseases such as diabetes, asthma, and chronic obstructive pulmonary disease (COPD) can be monitored and diagnosed non-invasively.

For example, Wang and others developed a biodegradable water-soluble triboelectric nanogenerator (WS-TENG) [96], made using recycled biodegradable paper and water-soluble graphite electrodes. By extracting cellulose nanocrystals (CNCs) from the paper and mixing them with methyl cellulose (MC), a CNC/MC composite film was formed, serving as the positive electrode material. This cost-effective, lightweight, and biodegradable WS-TENG has been innovated as a bandage sensor that can be readily dissolved in water. Contrasting with conventional gauze sensors, the water-insoluble components of this cellulose-based apparatus can be facilely isolated from water for subsequent reuse. The variations in the output voltage of the WS-TENG are capable of precisely distinguishing between various respiratory states, rendering it adequate for employment as a real-time physiological signal-monitoring sensor. As a fully water-soluble device, it has the potential to act as a sophisticated medical sensor, thereby broadening the scope of TENG applications in health monitoring.

Kim and colleagues developed a new type of highly elastic and self-healing hydrogel conductor (CCDHG) [42], illustrated in Figure 7a, using all-marine biomaterials, including catechol, CTS, and diatom. The CCDHG was employed in fabricating a stretchable TENG, which exhibited an open-circuit voltage of up to 110 V, a short-circuit current of 3.8 μA, and an instantaneous power density of 29.8 mW/m^2^, as shown in Figure 7b. Furthermore, it served as a self-powered tremor sensor, attached to the skin, for monitoring the health status of Parkinson’s Disease (PD) patients, as depicted in Figure 7c.

Liu developed a CTS and zinc oxide (ZnO)-based TENG for room temperature acetone detection with excellent humidity tolerance [97], attributed to the numerous hydrogen bonds formed between CTS and ZnO. The bilayer structure of CTS and ZnO not only acts as the sensing layer but also as the triboelectric layer, achieving high sensitivity and low detection limit for acetone, particularly suitable for the breath diagnosis of diabetes.
Figure 7(**a**) Schematic representation of the CCDHG−TENG apparatus; (**b**) the electrical output metrics of the CCDHG−TENG; (**c**) the schematic depiction of the application of the CCDHG−TENG in Parkinson’s disease monitoring; reprinted with permission from Ref. [42]. Copyright 2021, Elsevier. (**d**) The production process diagram for the FG−TENG, inspired by leaf microstructures; (**e**) the generated output voltage and current of the LMFG−TENG across varied resistances; (**f**) the application of LMFG−TENG on the human body to track movements such as walking, running, jumping, leg swinging, and vocal cord vibrations; reprinted with permission from Ref. [98]. Copyright 2023, Elsevier. (**g**) The structural blueprint of the P−TENG; (**h**) the performance metrics of the P−TENG; (**i**) the rvaluation of the P−TENG’s antibacterial effectiveness against *Escherichia coli* and *Staphylococcus aureus*. Reprinted with permission from Ref. [99]. Copyright 2021, MDPI.
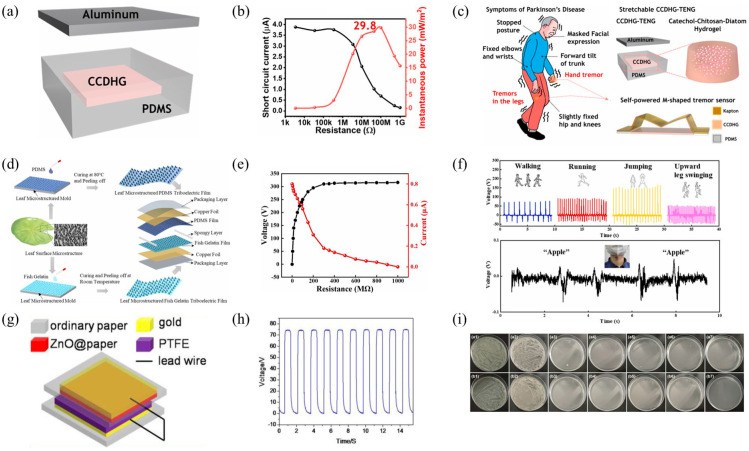


#### 4.2.2. Motion Tracking

Biopolymers, with their unique biocompatibility, degradability, and flexibility, are increasingly becoming important materials for motion tracking. The application of biopolymers makes wearable devices lighter and more comfortable. For example, wearable sensors based on biopolymers can be integrated into sports clothing, wristbands, or insoles, monitoring real-time motion data such as step count, speed, movement trajectory, calorie consumption, and even muscle activity and posture. Through the real-time tracking and analysis of these data, individuals can better understand their physical conditions and formulate and adjust personal fitness plans, thereby achieving a healthier lifestyle. Moreover, due to the good biocompatibility of biopolymers, they can also be used to develop implantable sensors for the long-term monitoring of chronic disease patients. This reduces the side effects of in-body use, such as inflammation or foreign body reactions, and these materials can naturally degrade when no longer needed, reducing residual in the body.

For example, Nie and others fabricated an excellent-performance, moisture-resistant TENG based on CNF [100]. By a simple and eco-friendly method, CNF films underwent aminosilane modification, significantly enhancing the CNF surface’s positive charge density and hydrophobicity. This functional CNF-based TENG exhibited excellent output stability at an environmental humidity of 70%, and was able to respond to various human activities such as pressing, stretching, bending, and twisting, showing outstanding flexibility. More importantly, this TENG could monitor human motion states in a sweaty environment, providing new insights and possibilities for the application of self-powered wearable electronic devices.

Shi and colleagues developed a FG-based triboelectric nanogenerator (FG-TENG) [98], inspired by the microstructural design of natural plant leaves, as shown in Figure 7d. This FG-TENG achieved significant improvements in power generation, capable of reaching a maximum voltage of up to 320 V and a current output of 0.80 μA, while also demonstrating good environmental degradability, as illustrated in Figure 7e. Importantly, the device has been successfully applied to monitor body posture, as depicted in Figure 7f, showcasing its potential applications in the healthcare sector, especially in self-powered sensing and human activity monitoring.

#### 4.2.3. Anti-Mite and Antibacterial 

Natural biomaterials are one of the preferred materials for constructing antibacterial TENGs. For example, Lu and others used a hydrothermal method to synthesize CTS-hydroxyethyl cellulose-pectin (CHP) films, used as triboelectric materials [101]. At 11% CTS content, the TENG exhibited a superior current and voltage output, with efficient energy conversion performance and anti-mite antibacterial properties, providing a potential solution for TENG-based intelligent medical and health monitoring systems.

Lin and others manufactured a flexible paper-based triboelectric nanogenerator (P-TENG) pairing ZnO@paper with PTFE film [99], as depicted in Figure 7g. This P-TENG not only demonstrated high output performance but also exhibited antibacterial activity. Specifically, the output voltage and current of the P-TENG were 77 V and 0.17 μA, respectively, as shown in Figure 7h. The ZnO@paper showed excellent antibacterial activity against *Escherichia coli* (*E. coli*) and *Staphylococcus aureus* (*S. aureus*), indicating that the P-TENG could inhibit and kill bacteria during operation, as illustrated in Figure 7i. The results also indicated that ZnO could improve the surface roughness of cellulose paper, enhancing the output performance of the flexible P-TENG. Furthermore, the potential applications of the P-TENG-based pressure sensor in measuring human motion information were reported.

### 4.3. Environmental Monitoring

#### 4.3.1. Air Quality

Air quality monitoring is of significant importance for public health and urban environmental management. Sensors made from biopolymers can be used to detect various pollutants in the air, including but not limited to NO_2_, NH_3_, humidity, PM2.5, carbon dioxide, etc.

For example, Yang and others developed a self-powered NO_2_ gas sensor based on paper and In_2_O_3_/SnS_2_ composite materials, powered by a TENG, achieving a high-sensitivity detection of NO_2_ [102]. Utilizing paper as the friction material, the sensor showed rapid response and recovery to 50 ppm NO_2_ under room temperature and 43% relative humidity. Additionally, a self-powered alarm system based on this technology was constructed for the real-time monitoring of NO_2_ concentration in the environment, suitable for smart environmental monitoring.

Yang and others developed a self-powered gas sensor (PC-TENG) based on polyaniline (PANI)/commercial cellulose paper for the efficient monitoring of NH_3_ concentration at room temperature (Figure 8a) [44]. Utilizing the three-dimensional microporous structure of commercial nitrocellulose filter paper as a porous framework base, the in situ polymerization of PANI nanoprotrusions formed a fine PANI hierarchical structure with excellent gas permeability and abundant NH_3_ adsorption sites, as illustrated in Figure 8b. Exhibiting exceptional NH_3_ detection capabilities, this PC-TENG sensor, shown in Figure 8c, reached a lower detection limit of 100 ppb and could detect levels up to 500 ppm. Its sensitivity, peaking at 45.41% ppm^−1^ within the 0.1 to 1 ppm range, stands as the highest for trace NH_3_ among current self-powered NH_3_ sensors. This innovation, highlighted at the outset, merges environmental sustainability with effective monitoring, making it appropriate for broad production and application.

Zheng and others developed an environmentally friendly and multifunctional starch-based triboelectric nanogenerator (S-TENG) [103]. The S-TENG showed an open-circuit voltage of 151.4 V and short-circuit current of 47.1 µA, not only able to drive and smartly control electronic devices but also effectively harvest energy from human motion and wind. Notably, the output of S-TENG increased instead of decreasing with the increase in environmental humidity, showing an unusual performance improvement. Within a relative humidity range of 20% to 80%, the S-TENG could act as a sensitive self-powered humidity sensor. This work paves the way for the mass production of multifunctional biomaterial-based TENGs and their practical application in self-powered sensing.
Figure 8(**a**) Schematic diagram of the TENG device based on PANI/cellulose paper; (**b**) the schematic illustration of the PANI/nitrocellulose layered structure; (**c**) the sensitivity comparison of the PC−TENG for NH_3_ with previously reported works; reprinted with permission from Ref. [44]. Copyright 2023, the Royal Society of Chemistry. (**d**) the design of the sensing system based on FF−TENG and its working mechanism diagram; (**e**) the output performance of wool fibers of different sizes; (**f**) wave sensor based on FF−TENG; reprinted with permission from Ref. [104]. Copyright 2023, Elsevier. (**g**) The schematic diagram of the SFC−TENG device; (**h**) the output power diagram of the SFC−TENG; (**i**) the application diagram of the SFC−TENG in agricultural production; reprinted with permission from Ref. [43]. Copyright 2022, Elsevier. (**j**) The preparation flowchart of the CNF−BP−PA film for TENG; (**k**) the process diagram of the FR-TENG for flame detection alarm; (**l**) the flowchart of the FR−TENG for temperature detection alarm. Reprinted with permission from Ref. [37]. Copyright 2022, Elsevier.
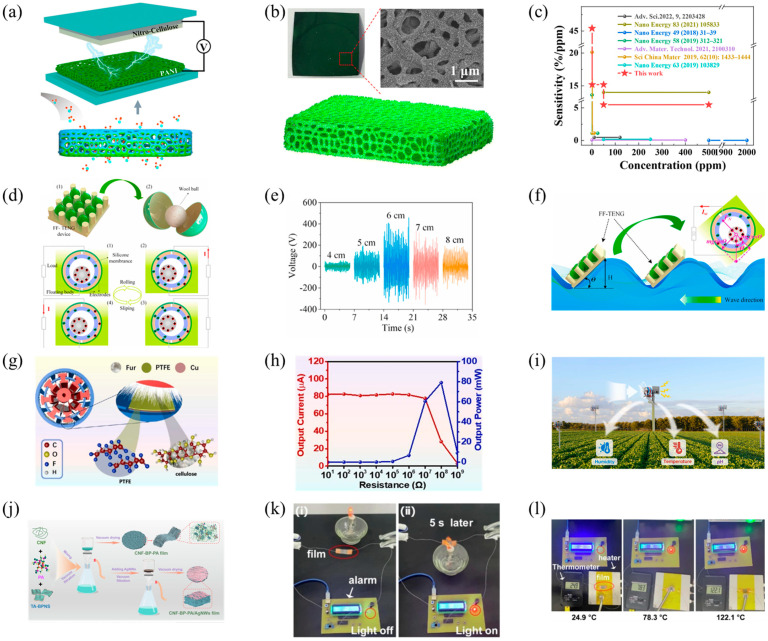


#### 4.3.2. Marine Accident

The real-time monitoring of the marine environment is crucial for preventing maritime accidents. Xia and their team introduced a novel freely floating structure triboelectric nanogenerator (FF-TENG) [104], specifically designed for environmental monitoring around marine wind turbines (Figure 8d). The FF-TENG employs an integrated structure of spheres and floaters, combined with lightweight wool balls as the triboelectric material, to efficiently capture and convert wave energy. Thanks to this unique design, the FF-TENG can adapt to ocean waves, generating up to 462 V and 15.5 μA of current, while maintaining good durability and stable energy output, as shown in Figure 8e. Not only does the device harvest wave energy, but it also serves as a self-powered wave sensor for the real-time monitoring of marine environmental changes, such as wave frequency and amplitude, offering an effective solution for the environmental monitoring of offshore wind turbines, as demonstrated in Figure 8f. This research provides an innovative approach to marine environmental monitoring, integrating energy harvesting and sensing into one solution.

#### 4.3.3. Agricultural Production

Self-powered wireless sensors have broad application prospects in the real-time monitoring of agricultural environments. Men and others developed a cotton cellulose-based triboelectric nanogenerator (SFC-TENG) for wind energy collection in smart agriculture (Figure 8g) [43]. The device utilizes an innovative dual rotor/stator structure, enhancing the energy conversion efficiency and durability. The SFC-TENG demonstrates an exceptional electrical output performance, including a high open-circuit voltage of 2500 V, a short-circuit current of 85 μA, and a maximum output power of 80 mW, as depicted in Figure 8h. It has been successfully applied in smart agriculture for nighttime lighting, pH value monitoring, and temperature and humidity control, as shown in Figure 8i, showcasing its potential as a clean and sustainable energy solution in agricultural technology.

#### 4.3.4. Urban Safety

##### Fire Alarm

Developing a flexible TENG with temperature-responsive and fire-warning capabilities is a challenging task, especially when considering the use of natural biopolymers as materials. Such a TENG needs to find a balance between physical properties and environmental adaptability while ensuring sufficient sensitivity to temperature changes. For example, Wang et al. fabricated a flexible, flame-retardant triboelectric nanogenerator (FR-TENG) utilizing materials such as cellulose, black phosphorus, and phytic acid [37], depicted in Figure 8j, featuring high-temperature responsiveness and fire warning capabilities. This FR-TENG exhibited exceptional fire resistance and flame-retardant properties. Most notably, the FR-TENG demonstrated extremely high thermal response sensitivity within the temperature range of 35–150 °C, capable of rapidly responding to temperature changes within 5 s, making it suitable for early fire detection, as shown in Figure 8k,l. This work provides new insights into developing temperature monitoring and early fire warning systems.

##### Traffic Warning

Future urban planning necessitates the integration of self-powered and sustainable traffic monitoring systems. Li et al. introduced an all-weather, self-powered intelligent traffic monitoring system, employing a polyvinyl alcohol-polyacrylamide/tannic acid-modified cellulose nanocrystal double network hydrogel (PPC) [105]. The system includes self-healing piezoresistive sensors and TENG, capable of the real-time monitoring of driver status and vehicle conditions. Due to its excellent self-healing performance and stable environmental adaptability, the system can operate across a wide range of temperatures, timely detecting vehicle speed and weight, assessing traffic accident responsibilities, and preventing accidents caused by fatigued driving. Furthermore, replacing the water component in PPC with glycerol further ensures its stable operation under various climatic conditions. This work provides new perspectives and technical support for the safety assurance and smart traffic development of future cities.

##### Smart Home

Hao and others demonstrated a wood-based triboelectric nanogenerator (W-TENG) made of natural New Zealand pine and PTFE [106], which not only can convert mechanical energy into electrical energy through human walking actions but also can be used as a self-powered switch sensor. As a lighting switch, W-TENG effectively reduces unnecessary energy consumption. Moreover, W-TENG can also be applied to security alarm systems, triggering alarms by monitoring specific activities or movements, thus achieving security monitoring and energy self-sufficiency without the need for an external power source, providing new solutions for smart homes and safety fields.

Most of the polymer materials used in manufacturing TENGs are non-biodegradable, leading to plastic pollution and potentially fatal effects on aquatic life. In the United States, the average person consumes approximately 39,000 fibers per day in drinking water [107]. Hence, there is a significant demand for biodegradable alternatives. Despite various TENGs made from both natural and synthetic polymer materials proving to be clean, cost-effective, and sustainable energy harvesters for self-powered electronic devices across various applications such as sensors, biomedical instruments, and smart home controllers, the adoption of new natural polymers for TENGs with mechanical durability and high power efficiency still faces severe environmental constraints, including humidity, temperature, electromagnetic interference, mechanical flexibility, transparency, breathability, hydrophobicity, and acoustics. For instance, exposure to humid conditions or repetitive external forces can result in the mechanical degradation of eco-friendly TENGs when using naturally sourced biopolymers. Hence, further research is warranted to explore novel biopolymer materials capable of overcoming the durability and performance limitations of existing eco-friendly TENGs.

Table 1 provides a detailed summary of various biopolymer-based TENGs, including their synthesis methods, materials, dimensions, triboelectric outputs, and main application areas. These materials demonstrate potential applications in natural resource collection, human health monitoring, and environmental surveillance, among others. They also exhibit the characteristics of biocompatibility and degradability, making them suitable for creating comfortable and portable wearable devices.

## 5. Conclusions and Prospect

Compared to conventional polymer materials, biopolymer materials are not only non-toxic, biocompatible, and biodegradable, but most importantly, they are abundantly available in nature and often possess higher surface areas. These are key factors influencing the output performance of TENGs. As indicated in Table 1, some BP-TENGs are comparable to existing inorganic-based TENGs, making them suitable for self-powered electronic products, thus demonstrating their potential as performance-enhancing triboelectric electric materials. Additionally, BP-TENGs exhibit significant application value in fields such as medical health monitoring and environmental control, especially in harnessing mechanical energy from natural and biological sources for autonomous, continuous, and reliable operation. Although there are still unresolved issues, the challenges related to the output performance of BP-TENGs are surmountable through strategic material selection, precise structural design, chemical modification, and the optimization of energy management systems. Meanwhile, the wear and degradation issues of biopolymers may affect the long-term stability and durability of BP-TENGs, but these challenges are expected to be addressed through advanced surface treatment technologies, cross-linking reinforcement strategies, the use of composite materials, and innovations in encapsulation techniques.

To push BP-TENG technology towards broader applications in energy harvesting, healthcare, and environmental monitoring, this study highlights several key challenges and issues that future research should focus on. These include improving energy conversion efficiency, extending device lifespan, enhancing system environmental stability, and ensuring the biocompatibility and sustainability of biopolymers. By addressing these issues, BP-TENG technology has the potential to revolutionize its applications in self-powered sensors, wearable devices, and environmental energy harvesting, thereby leading a new round of technological innovation and sustainable development strategies:

### 5.1. Output Performance

The energy conversion efficiency of current BP-TENGs still has room for improvement compared to traditional energy-harvesting technologies, necessitating improvements in material properties and device structure. Strategies to enhance BP-TENG performance include selecting suitable materials, the micro/nano-structuring of surfaces, and chemical modifications, such as nitration and amination, which can alter the surface potential of biopolymers and, thereby, the charging states during contact electrification. However, most methods are both complex and costly to implement. Additionally, there is a lack of in-depth study on the intrinsic multi-level structure of biopolymers and their triboelectric performance in BP-TENG reports. In the future, more material modification techniques will be introduced into the development of BP-TENGs to enhance their electrical performance and mechanical quality. These advancements will provide greater possibilities and challenges for BP-TENGs in driving electronic devices and achieving effective electrical stimulation. Overall, BP-TENGs are expected to find broader applications in the future.

### 5.2. Stability and Durability

Although biopolymers offer advantages in biocompatibility and degradability, they often face the issues of declining mechanical performance, rapid biodegradation, or increased environmental sensitivity in long-term applications. Especially under the conditions of repetitive mechanical stress, high humidity, or temperature changes, the structure of biopolymers might alter, leading to a decreased triboelectric performance. Moreover, biopolymers may be sensitive to microbial activities, which could accelerate material degradation in certain environments, thereby shortening the effective lifespan of BP-TENG devices. These challenges require researchers to innovate in material selection and design and to enhance the stability and durability of biopolymers through surface treatment, cross-linking reinforcement, composite materials, and the development of encapsulation techniques. Addressing these issues is crucial for the widespread application of BP-TENG technology in energy harvesting, self-powered sensors, and sustainable electronic devices. Therefore, developing BP-TENG systems with high stability and durability represents a major challenge in the field, as well as a significant opportunity for research and technological innovation.

### 5.3. Scalable Production and Standardization

Despite the great potential of BP-TENGs in energy harvesting and sensor applications, their large-scale production and widespread application still face many obstacles. The diverse sources and varying performances of biopolymers lead to an increased complexity in production processes and variability in final product performance. Moreover, the processing and biocompatibility treatment of biopolymers require delicate and costly approaches, further limiting the scalability of BP-TENGs. The challenges in standardization should not be overlooked either, as there is currently a lack of unified standards to evaluate and verify the performance and safety of different BP-TENG products. To overcome these challenges, efficient and economical production processes need to be developed, and comprehensive standards and testing regulations should be established.

### 5.4. Controlled Degradation

BP-TENGs typically leverage biopolymers to enhance their biocompatibility and biodegradability. To date, the degradation of BP-TENGs has been regulated through various physical and chemical approaches, including methanol treatment and infrared control, but these do not allow for the precise control of material degradation rates in natural or biological environments. This involves complex materials science issues, including how to ensure that materials safely degrade within a predetermined time while maintaining good energy collection performance. Additionally, degradation rates are influenced by various factors, including the chemical composition, structure, and external environmental conditions (such as temperature, pH value, and enzyme activity), making controlled degradation more complex. Moreover, the products of complete degradation need to be non-toxic and harmless to avoid adverse effects on the environment or human health. Although these challenges are daunting, they also offer broad opportunities for researchers to explore new materials, technologies, and degradation mechanisms, thereby advancing the further application of BP-TENG technology in sustainable development and biomedicine.

### 5.5. Integration and Compatibility

Integration and compatibility are crucial for the practicality and widespread application of BP-TENGs. This involves not only the multifunctionality of BP-TENG devices and their ability to work in synergy with other systems but also their adaptability to different application environments. Due to the unique properties of biopolymers, such as biodegradability and biocompatibility, they may face material matching and interface compatibility issues when integrating with existing technologies and electronic devices. Additionally, the energy output characteristics of BP-TENGs need to match the energy demands of backend electronic systems, requiring the development of efficient energy conversion and management strategies to ensure continuous and stable power supply. Through successful integration, BP-TENGs are expected to play a significant role in wearable devices, self-powered sensor networks, biomedical applications, and environmental monitoring in the future. Moreover, this integration will also drive the development of a new generation of eco-friendly and self-sustaining energy solutions, providing strong support for the integration of green energy conversion and intelligent technologies.

## Figures and Tables

**Figure 1 polymers-16-01304-f001:**
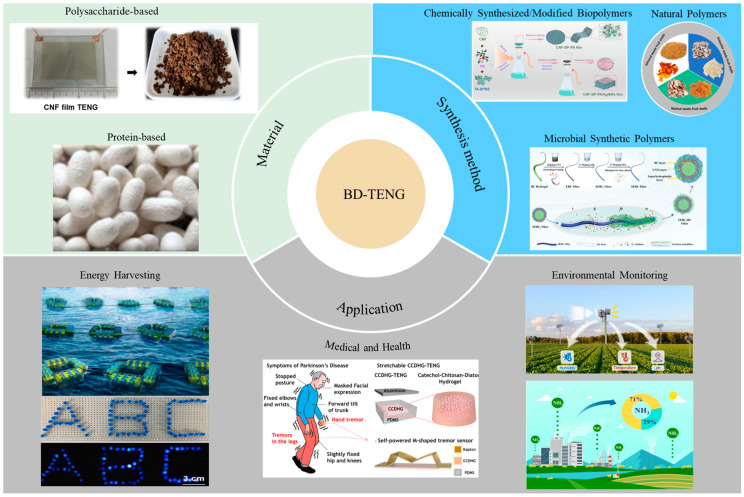
Recent progress of TENG based on biopolymer, including polysaccharide, reprinted with permission from Ref. [35]. Copyright 2016, Elsevier. Protein, reprinted with permission from Ref. [36]. Copyright 2023, Elsevier. Chemical synthetic source, reprinted with permission from Ref. [37]. Copyright 2022, Elsevier. Natural source, reprinted with permission from Ref. [38]. Copyright 2022, Elsevier. Microbial source, reprinted with permission from Ref. [39]. Copyright 2023, Wiley Online Library. Energy harvesting, reprinted with permission from Ref. [40]. Copyright 2021, Elsevier. Reprinted with permission from Ref. [41]. Copyright 2023, Elsevier. Medical and health, reprinted with permission from Ref. [42]. Copyright 2021, Elsevier. Environmental monitoring, reprinted with permission from Ref. [43]. Copyright 2022, Elsevier. Reprinted with permission from Ref. [44]. Copyright 2023, the Royal Society of Chemistry.

**Figure 2 polymers-16-01304-f002:**
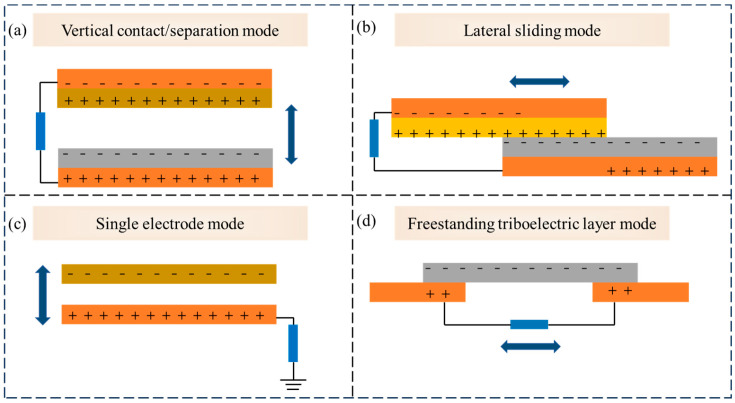
The four core operational configurations of TENG encompass (**a**) the vertical contact/separation mode; (**b**) the lateral−sliding mode; (**c**) the single−electrode mode; and (**d**) the free−standing triboelectric layer mode.

**Figure 3 polymers-16-01304-f003:**
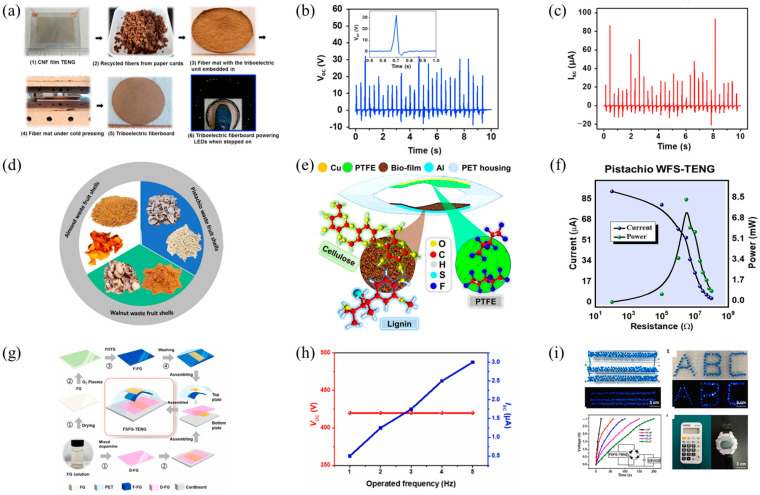
(**a**) Manufacturing flowchart for TENG fiber board; (**b**) V_OC_ and (**c**) I_SC_ outputs when TENG fiber board is repeatedly stepped on by a person of normal weight; reprinted with permission from Ref. [35]. Copyright 2016, Elsevier. (**d**) A photograph of the lignin powder derived from discarded biomaterials; (**e**) the schematic diagram of the TENG device based on wood fiber substrates; (**f**) the output power of the Pi-WFS-based device under different external load resistances, ranging from 100 Ω to 100 MΩ; reprinted with permission from Ref. [38]. Copyright 2022, Elsevier. (**g**) Manufacturing flowchart for the FSFG−TENG; (**h**) the output performance of the TENG across different frequencies; (**i**) FSFG−TENG powering various electronic devices. Reprinted with permission from Ref. [40]. Copyright 2021, Elsevier.

**Figure 6 polymers-16-01304-f006:**
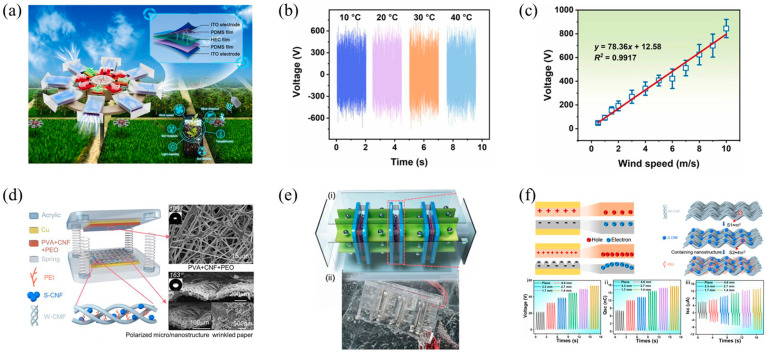
(**a**) Schematic of the HEC−TENG for self-powered wind direction and velocity monitoring; (**b**) the output voltage of the HEC−TENG at different temperatures, with a wind speed of 7 m/s; (**c**) the linear fitting graph of wind speed and wind pressure measured by an array of 8 CE−TENGs; reprinted with permission from Ref. [95]. Copyright 2022, Elsevier. (**d**) The schematic diagram of the W−TENG device; (**e**) the schematic and photograph of a G−TENG array with three units; (**f**) the schematic illustration of the relationship between triboelectric output and surface roughness, with wrinkling range from 1.4 to 4.6 mm. Reprinted with permission from Ref. [41]. Copyright 2023, Elsevier.

**Table 1 polymers-16-01304-t001:** Summary of the performance and application of the BP-TENG with different triboelectric materials.

Synthesis Method	Material	Type	Size	Electrical Output	Application	Ref.
Natural	Silk	Proteins	4 × 7 cm^2^	2 V/4.3 mW·m^−2^	Energy harvesting	[34]
Natural	SF	Proteins	2 × 4 cm^2^	68 V/5.78 μA	Drive microdevice	[63]
Natural	SF	Proteins	2 × 1 cm^2^	172 V/8.5 μA	Drive microdevice	[36]
Natural	SF	Proteins	2 × 2 cm^2^	∼50 V/∼3 μA	Intelligent vehicle	[64]
Natural	Spider silk protein	Proteins	6 × 8 cm^2^	≈2.6 kV/≈0.48 mA	Implantable anti-bacterial patch	[66]
Natural	RP	Proteins	6 cm (diameter)	∼70 V/∼2.6 μA	Medical devices	[65]
Natural	Cellulose	Polysaccharides	2 × 2 cm^2^	~96 V/130 mW·m^−2^	E-skin	[58]
Natural	Cellulose	Polysaccharides	1 × 1 cm^2^	~30 V/~90 μA	Power board	[61]
Natural	Lignin	Polysaccharides	6.5 × 6.5 cm^2^	1.04 V/cm^2^/3.96 nA/cm^2^	Biomedical devices	[59]
Natural	Lignin	Polysaccharides	4.5 × 4.5 cm^2^	700V/95 μA	Energy harvesting	[38]
Natural	Starch	Polysaccharides	2 × 2 cm^2^	22 V	Biomedical devices	[60]
Natural	Gelatin	Proteins	3 × 3 cm^2^	500 V/4 μA	Wearable devices	[40]
Natural	Paper	Polysaccharides	6 × 3 cm^2^	180 V/20 μA	Health care	[96]
Microbial Synthetic	PHB	—	8 mm (diameter)	25.6 V/cm^−2^/550.2 nA·cm^−2^	Athletic monitoring	[72]
Microbial Synthetic	BC	Polysaccharides	6 × 6 cm^2^	29 V/0.6 μA	Wearable devices	[73]
Microbial Synthetic	BC	Polysaccharides	20 × 0.5 cm^2^	266.0 V/5.9 µA	Athletic monitoring	[39]
Microbial Synthetic	γ-glycine	—	3 × 3 cm^2^	81 V/121 μA	Energy harvesting	[74]
Microbial Synthetic	Aspartic acid	—	2.5 × 2.5 cm^2^	200 V/6 μA	Gas sensor	[75]
Microbial Synthetic	CTS	Polysaccharides	3 × 4 cm^2^	121 V/15 µA	Energy harvesting	[76]
Microbial Synthetic	CTS	Polysaccharides	1 × 1 cm^2^	106.04 ± 2.3 V	Energy harvesting	[77]
Microbial Synthetic	SA	Polysaccharides	5 × 5 cm^2^	33 V/150 nA	Energy harvesting	[78]
Microbial Synthetic	SA	Polysaccharides	5 × 5 cm^2^	53 V/18 nC	Wearable devices	[79]
Microbial Synthetic	SA	Polysaccharides	3 × 3 cm^2^	629 V/40.16 μA	Self-powered sensing array	[80]
Microbial Synthetic	κ-Carrageenan-agar	Polysaccharides	3 × 3 cm^2^	0.45 mA·m^−2^/0.15 mW·m^−2^	Energy harvesting	[83]
Chemically Synthesized	PLA	—	8 cm (Diameter)	395 V/28 μA	Drive microdevice	[85]
Chemically Synthesized	PCL	—	3×3 cm^2^	800 V/30 μA	Self-healing	[86]
Chemically Synthesized	PCL	—	16 cm^2^	~1.1V/~45 nA	Wearable devices	[87]
Chemically Synthesized	CD	Polysaccharides	2 × 2 cm^2^	152 V/1.2 μA	Energy harvesting	[88]
Chemical Modification	CA	Polysaccharides	20 × 3 × 0.08 mm^3^	103.2 V/7.93 mA·m^−2^	Motion tracking and wind speed	[92]
Chemical Modification	CA	Polysaccharides	2 × 1 cm^2^	~400 V/~3 mA/m^2^	Vibration sensor	[89]
Chemical Modification	PU	—	2 × 2 cm^2^	120 V/1.2 μA	Self-healing	[93]
Chemical Modification	Cellulose paper	Polysaccharides	4 × 4 cm^2^	126 V/11.4 μA	Energy harvesting	[94]
Chemical Modification	Hydroxyethyl cellulose	Polysaccharides	5 × 1 × 7 mm^3^	584 V/41 μA	Agricultural production	[95]
Chemical Modification	Cellulose	Polysaccharides	2 × 4 cm^2^	195 V/13.4 μA	Wearable devices	[100]

“—” in the table indicates that the data were not documented in the research.

## Data Availability

Data are available on request from the authors.

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
