# Peer review of "Biopolymer Materials in Triboelectric Nanogenerators: A Review"

_polymers, 2024, doi:10.3390/polym16101304_

Round 1

Reviewer 1 Report

Comments and Suggestions for Authors

see file

Comments on the Quality of English Language

Some editing of English required

Author Response

1.Such important points as the fundamental physics of TENGs, i.e. origin of triboelectrification, modes of TENGs, etc. not presented.

Response: We indeed appreciate your valuable comments. According to your instructions, the basic physics of triboelectrification, the origin of triboelectrification and the working mode of triboelectrification have been modified in the second part.

2. There are also no critical conclusions for each biopolymer material and device

considered in the review.

Response: Thank for your suggestions. The conclusions for biopolymer materials have been revised in Parts 3 and 4.

3.The review is overloaded with many general propositions without clear conclusions. I strongly recommend improving the style of this manuscript, especially in the final part.

Response: Thank for your valuable comments. The last part has been modified.

Reviewer 2 Report

Comments and Suggestions for Authors

The review represents a well-balanced paper covering basic work principles of triboelectric nanogenerators (TENG), recent progress in utilization of various types of biopolymers for TENG, TENG applications and valuable discussion on pros and cons of TENG towards upscaling and industrial use. The article is based on up-to-date materials and can be recommended to publication after minor revision. Please, consider the below presented comments.

1.    In the abstract, sentence “The development of these TENG devices has sparked a green revolution in energy technology, reducing reliance on fossil fuels and mitigating potential environmental pollution from metals and non-degradable plastic materials.” overestimates potential of TENG. Currently, they produce power at relatively low levels. Thus, TENGs are used for a limited range of applications and their total impact on the reduction of fossil fuel consumption and decrease of pollutions is minor. It is recommended to reformulate the sentence.

2.    Section 3 describes various types of biopolymers used for TENG. This description is presented as a simple listing of individual publications without concluding on performances of one on another type. How the performances of biopolymers are comparable with ordinary synthetic polymers? It would be nice to provide some comprehensive analysis and summarise at the end.

There is a number of small technical imperfections.

Line 17. In phrase “friction-based nanogenerators (TENGs)” the shortcut does not correspond to the term. It should be triboelectric nanogenerators.

Lines 26 and 67. In shortcut “BP-TENG”, BP is not explained for the first time used, while explained only in line 74. It should be corrected.

Reference 35 does not cite first publication about TENG. It should be paper published in Nano Lett. 2012.

Figure 1 is not mentioned anywhere in the text. It also looks like some of the panels of this figure are adopted from other publications. In this case, appropriate citations (covered by permissions) should be provided in the caption.

Line 178. Subtitle “Protein-based” is better to extend to “Protein-based BP” or similar.

Lines 325 and 359. Subtitles “Fungal Synthesis” and “Algae Synthesis” also sound fragmental. It is recommended to extend the titles.

Lines 345, 452, 463, 484 and more. Either leave triboelectric nanogenerator or TENG.

Similar to TENG, some other shortcuts are explained a number of times. For example, PLA on p. 11 and 12 or CA on p. 13. Some shortcuts, like CH and AL, are introduced but not used afterwards. Some shortcuts, like SE-TENG on p. 15, do not agree with the terms, “crack-effect based triboelectric nanogenerator”. There are many such cases.

It is recommended to make extensive proof reading regarding the shortcuts.

Comments on the Quality of English Language

English is fine.

Author Response

1.In the abstract, sentence “The development of these TENG devices has sparked a green re volution in energy technology, reducing reliance on fossil fuels and mitigating potential environmental pollution from metals and non-degradable plastic materials.” overestimates potential of TENG. Currently, they produce power at relatively low levels. Thus, TENGs are used for a limited range of applications and their total impact on the reduction of fossil fuel consumption and decrease of pollutions is minor. It is recommended to reformulate the sentence.

Response: We indeed appreciate your valuable comments. According to your instructions, this paragraph has been amended.

2.Section 3 describes various types of biopolymers used for TENG. This description is presented as a simple listing of individual publications without concluding on performances of one on another type. How the performances of biopolymers are comparable with ordinary synthetic polymers? It would be nice to provide some comprehensive analysis and summarise at the end.

Response: Thank for your suggestions. We have made changes in the last paragraph.

3.There is a number of small technical imperfections.

3.1 Line 17. In phrase “friction-based nanogenerators (TENGs)” the shortcut does not correspond to the term. It should be triboelectric nanogenerators.

Response: Thank for your valuable comments. "friction based nanogenerators (TENGs)" has been changed to "triboelectric nanogenerators (TENGs)"

3.2 Lines 26 and 67. In shortcut “BP-TENG”, BP is not explained for the first time used, while explained only in line 74. It should be corrected.

Response: Thank for your suggestions. BP-TENGs is already marked on line 23.

3.3 Reference 35 does not cite first publication about TENG. It should be paper published in Nano Lett. 2012.

Response: Thank for your valuable comments. The article on Nano Lett has been replaced in document 35.

3.4 Figure 1 is not mentioned anywhere in the text. It also looks like some of the panels of this figure are adopted from other publications. In this case, appropriate citations (covered by permissions) should be provided in the caption.

Response: Thank for your suggestions. Figure 1 has already been mentioned in the third paragraph of the introduction to the article, and the reference and authorization in the Figure 1 title have been added.

3.5 Line 178. Subtitle “Protein-based” is better to extend to “Protein-based BP” or similar.

Response: Thanks for your professional review work on our manuscript. According to your instructions, subheadings such as "Protein-based" have been changed.

3.6 Lines 325 and 359. Subtitles “Fungal Synthesis” and “Algae Synthesis” also sound fragmental. It is recommended to extend the titles.

Response: Thank for your suggestions. The corresponding title has been expanded.

3.7 Lines 345, 452, 463, 484 and more. Either leave triboelectric nanogenerator or TENG.

Response: Thank for your comments. The corresponding place has been changed to TENG

3.8 Similar to TENG, some other shortcuts are explained a number of times. For example, PLA on p. 11 and 12 or CA on p. 13. Some shortcuts, like CH and AL, are introduced but not used afterwards. Some shortcuts, like SE-TENG on p. 15, do not agree with the terms, “crack-effect based triboelectric nanogenerator”. There are many such cases.

It is recommended to make extensive proof reading regarding the shortcuts.

Response: Thank for your suggestions. Abbreviations such as PLA, CA, CH, AL, PU, SE-TENG have been changed.

Round 2

Reviewer 1 Report

Comments and Suggestions for Authors

Comments on the Quality of English Language

I suggest minor editing of English